# User Privacy Protection via Windows Registry Hooking and Runtime Encryption

**DOI:** 10.3390/s24165106

**Published:** 2024-08-06

**Authors:** Edward L. Amoruso, Richard Leinecker, Cliff C. Zou

**Affiliations:** 1Department of Electrical and Computer Engineering, University of Central Florida, Orlando, FL 32816, USA; edward.amoruso@ucf.edu; 2Department of Computer Science, University of Central Florida, Orlando, FL 32816, USA; richard.leinecker@ucf.edu

**Keywords:** Windows registry, hooking, privacy protection, runtime encryption

## Abstract

The Windows registry contains a plethora of information in a hierarchical database. It includes system-wide settings, user preferences, installed programs, and recently accessed files and maintains timestamps that can be used to construct a detailed timeline of user activities. However, these data are unencrypted and thus vulnerable to exploitation by malicious actors who gain access to this repository. To address this security and privacy concern, we propose a novel approach that efficiently encrypts and decrypts sensitive registry data in real time. Our developed proof-of-concept program intercepts interactions between the registry’s application programming interfaces (APIs) and other Windows applications using an advanced hooking technique. This enables the proposed system to be transparent to users without requiring any changes to the operating system or installed software. Our approach also implements the data protection API (DPAPI) developed by Microsoft to securely manage each user’s encryption key. Ultimately, our research provides an enhanced security and privacy framework for the Windows registry, effectively fortifying the registry against security and privacy threats while maintaining its accessibility to legitimate users and applications.

## 1. Introduction

The Windows registry, hereinafter referred to as the registry, is a fundamental component of the Windows operating system (OS) that stores a vast amount of configuration information and settings for various applications. The registry contains configuration data that are essential for an enhanced user experience, including profiles for each user, installed applications, document types, folder and icon settings, hardware configurations, and port references [1]. As a result, during OS operation, Windows frequently accesses and modifies these settings to ensure the proper functioning of many applications.

Numerous techniques exist for extracting data from the registry. Two of the most common tools included with the OS are RegEdit.exe, which offers a graphical user interface and is often used for browsing and editing registry keys, and Reg.exe, which operates via a command-line interface and is commonly used for scripting and automating registry tasks. These tools are frequently employed to diagnose issues and optimize system configuration settings. In addition to these, there are several internet-based programs freely available such as RegScanner [2], Registry Explorer [3], and Registry Viewer [4]. Moreover, Windows PowerShell (a powerful scripting language) can also access and parse registry data. Lastly, Microsoft offers developers a comprehensive and thoroughly documented application programming interface (API) for interacting with registry records [5]. Table 1 shows the list of abbreviations used in this paper.

While the registry offers invaluable functionality, it simultaneously introduces substantial risks, particularly pertaining to privacy and system security. The data encased in the registry have the potential to unveil sensitive insights into a user’s activities and professional responsibilities, rendering them susceptible to privacy breaches. Furthermore, the registry unveils the user’s system security configuration, potentially serving as an exploitable entry point for more advanced assailants.

Research has shown that an application developed using native Windows APIs can construct a user’s digital footprint by collecting registry information [6]. This footprint includes sensitive data from frequently used documents, files, and applications, such as Word, Excel, and PowerPoint in Microsoft Office 365. For instance, it may reveal specific details about confidential files, such as payroll records, prototype design documents, or sales commissions reports, which are identified by their corresponding names and locations.

The ease with which sensitive data can be accessed from the registry (e.g., using RegEdit.exe or PowerShell) poses a significant security threat. To address this vulnerability, we propose a novel approach that involves the real-time encryption and decryption of registry data. This proactive defense strategy ensures that critical information remains protected even in the event of unauthorized access to registry data.

A key benefit of our proposed solution is its ability to mitigate privacy risks in real time. For example, when a user works on a confidential document, the system timestamps the activity and stores the document’s name in the registry—a potential vulnerability for the user’s privacy if such data are exposed to other users or attackers. However, with our runtime encryption-based approach, even if an attacker gains access to the registry, they will be unable to crack the encrypted data and exploit this information. Our solution prevents unauthorized parties from gathering insights into a user’s activities, such as the types of documents, spreadsheets, and presentations they work with. 

The key contributions of this paper are as follows:Introducing a new paradigm of OS-based runtime function hooking and encryption of the registry of user-specific data. As far as we know, this is the first research on runtime registry encryption.Ensuring through our advanced hooking technique that the proposed registry encryption approach is transparent to both users and applications with minimal impact on system performance.Preventing malware and unauthorized actors from reading user’s confidential information in the registry.Providing a customized ready-to-use application capable of hooking and encrypting user-specific registry data.

## 2. Related Work

Throughout our research, we found no existing studies on the real-time encryption of user private information stored in the registry. However, several articles discussed techniques for securing registry information. These included the following:Disk encryption technologies: the encryption of entire disk volumes or partitions can help protect the registry from unauthorized access [7].Access-control lists (ACLs): the implementation of ACLs can restrict access to specific registry keys and values, ensuring that only authorized users or applications can modify or read-protected data [8].Anomaly detection tools: the utilization of detection tools can help identify suspicious registry modifications, enabling swift response and mitigation efforts [9].

Our research takes a more fine-grained approach to protecting private user data located in the registry, building on the promise of existing techniques. We explore runtime software-based encryption and decryption methods that target specific data within the registry, providing an additional layer of security for sensitive information. In the following, we will delve deeper into the first two techniques and discuss their limitations. The third technique, anomaly detection, adopts a reactive stance by continuously monitoring and blocking suspicious access attempts. While effective in responding to threats, this approach presents a significant challenge: ensuring comprehensive protection without encrypting the registry data.

### 2.1. Disk Encryption

The encryption of data offers strong protection for sensitive and privileged information. In full-disk encryption (FDE), a private key is used to encrypt the entire volume of a system. Several solutions offer this capability, including Microsoft’s BitLocker, VeraCrypt, and Symantec Encryption. All the mentioned solutions provide encryption for the entire disk, which helps protect registry information from data theft or exposure when a device is lost, stolen, or inappropriately decommissioned [10]. However, these solutions only protect the data when the user has turned off the machine. In other words, when the user logs into the machine, the data are unencrypted and accessible to any user’s running processes.

The registry stores its information in several system-level files on the disk. Implementing disk-level encryption on these files would require a substantial overhaul of the Microsoft OS, which is not a feasible solution for most customers. Additionally, in a multi-user environment, where multiple users have their own accounts on the same system, implementing file-level encryption on the registry files would lead to conflicts and disrupt normal system operation. For example, consider two users, Alice and Bob, who both want to encrypt the registry files with their own keys. If Alice encrypts the registry files with her key, Bob’s access to the registry would be restricted, as he would not have the decryption key and vice versa. To overcome these issues, both users would need to share the same encryption key, which would defeat the purpose of encryption to protect users’ privacy.

### 2.2. Access-Control Lists

Another method of securing registry data is the use of access-control lists (ACLs). Each registry key has a security descriptor that can be leveraged to configure access control for subkeys and their values. Common tools like Windows regedit and Group Policy Editor can be used to add permissions or access to certain registry values.

ACLs are a simpler and more efficient method of managing access control than relying solely on encryption algorithms [11]. This is because ACLs provide a straightforward way to control access, whereas encryption algorithms necessitate key management, introducing complexity and overhead. However, when it comes to the registry, it is essential to consider the dynamic nature of registry key creation and the associated data generated by specific applications. In such cases, ACLs can become overly complex and difficult to manage. Additionally, some applications may not be compatible with the registry access-control model, which can limit their ability to interact with the registry. Other issues, such as the risk of misconfiguration, registry bloat (which can degrade system performance), and integrity, could affect the effectiveness of using ACLs.

In our approach, we provide an alternative by using a technique to selectively encrypt specific registry keys, eliminating the continuous management and other possible issues associated with ACLs. With the use of transparent and adaptive encryption, this method becomes well suited for managing the constantly changing environment of registry keys and their associated data.

## 3. Threat Model

Our threat model examines the various ways in which an attacker might exploit access to the registry. While there are many potential threats in this area, our scenario involves a malicious user gaining unauthorized access to registry information, either by exploiting elevated privileges as an insider or by obtaining login credentials for another user account with registry access. With this level of access, the attacker could potentially extract substantial amounts of sensitive data, compromising the privacy and security of other users [12,13].

To prevent such unauthorized access, our system employs on-the-fly encryption and decryption of the registry data using advanced system-level hooking techniques in conjunction with an OS-managed encoding key. Specifically, each user is assigned a unique encryption key, allowing only authorized individuals to access their own sensitive registry information. This effectively blocks a malicious actor who obtains a particular user’s login credential from accessing other users’ sensitive data.

Furthermore, the encryption key of each user is securely stored and managed by the operating system’s security subsystem, making it inaccessible to other users or attackers’ applications even with elevated privileges. In essence, the OS seamlessly handles the encryption and decryption process, preventing any user from accessing another user’s registry data regardless of their access level (e.g., administrator, standard user, or power user).

## 4. Proposed Approach

To safeguard user privacy, we propose an approach that prevents unauthorized access to specific registry data. Our method involves hooking into the application responsible for accessing and storing data in the registry. We then encrypt the registry key data as they are written to the registry and decrypt them as they are read out. Figure 1 illustrates our proposed system architecture, highlighting the components involved in protecting user privacy.

This paper focuses on securing personal information stored by Microsoft Office applications (e.g., Word, Excel, and PowerPoint) in the registry. Our approach can be easily extended to protect other sensitive information in the registry created by other Windows software. The pseudo code for our proposed system is presented in Algorithm 1, which is illustrated in Figure 1.
**Algorithm 1: Pseudo code for the proposed scheme**Initialize pointers for registry functionsInitialize list of protected registry keysCreate pointers of registry functions to our functionsSet the parameters**while** (monitoring not equal to end) **do**       **if** (read registry key and key in protect list) **then**              **decrypt** (registry key data)              **if** (hash equals success) **then**                   **return** (unencrypted data)              **else**                   **return** (no-data)              **end**       **end**       **if** (write registry key and key in protect list) **then**             **encrypt** (registry key data)             **if** (hash equals success) **then**                  **return** (encrypted data)             **else**                  **return** (no-data)         **end**       **end****end**

To achieve registry protection for an application, we deploy Microsoft native APIs to capture and encrypt/decrypt the relevant registry information while maintaining application stability. Our goal is to target key registry values that contain each user’s private data, encrypting only the selected private registry information. In our prototype, we incorporated the following design features:Developed in C++ using native Windows APIs to enhance performance and support low-level operations, such as memory pointers. Microsoft recommends languages like C or C++ for these tasks due to their closer interaction with system-level APIs. Unlike higher-level languages (e.g., Python, Java, and Perl) which require an interpreter, C++ is compiled directly into machine code, providing additional efficiency.Highly customizable, allowing for the easy addition of new registry keys that require encryption and decryption. We create a comparison keyword table that is easily updated in our code, enabling the addition of more registry keys used by other programs for security protection.Support for both 32-bit and 64-bit applications, providing a comprehensive solution for various use cases. For example, if using the 32-bit version of Word, a 32-bit version of our software will be required due to the architecture of memory addressing.Transparent monitoring of the target application’s interaction with the registry, performing encryption and decryption when it detects a registry key name requiring protection. Our executable requires no installation; it only needs to be started with the target application that requires security protection.Use of an industry-proven cryptographic algorithm called the Advanced Encryption Standard (AES) for encryption and decryption. Considered widely adopted and approved by the U.S. National Security Agency (NSA) for its reliability and security.Key management performed by the Windows OS through its provided data protection API (DPAPI), delivering an additional layer of protection by hiding the keys from both applications and users. The DPAPI is transparent to the user, eliminating the danger of exposing the encryption key to any user or application [14].

### 4.1. Registry Modification

The interaction between applications and the registry is facilitated by APIs provided by the operating system [15]. However, Microsoft strongly cautions against direct modification of the registry file by applications, as this approach can compromise system stability. To ensure safe and reliable access to registry values, applications should instead utilize the Windows-provided API functions, which offer a controlled and stable interface for reading and writing registry data.

Our approach uses Microsoft’s Windows APIs (works on both Windows 10 or Windows 11) and the “Detours” software package (version 4.0.1) to intercept and manipulate registry data [16]. As shown in Figure 2, this technique enables us to modify specific registry values without affecting the target application’s functionality. This strategy is similar to a man-in-the-middle attack, in which our program intercepts and alters communication between the user’s application and the registry. By using this technique, we can ensure that changes are made to the registry data while maintaining transparency for the original Windows applications.

### 4.2. Securing with Encryption

The robustness of encryption algorithms and key management techniques are essential factors in protecting sensitive information in our proposed system. Considering the significance of these factors, we opted for Microsoft’s data protection API (DPAPI), a built-in feature in Windows 2000 and later versions [14]. The DPAPI leverages the Advanced Encryption Standard (AES), specifically the AES-256 key size, and supports the transparent (no user interface) encryption and decryption of data securely without the need to develop our own. Unlike other encryption APIs (CryptoAPI or Cryptography API), the DPAPI is designed to simplify the process of employing cryptographic techniques by removing the need for users to manage and store encryption keys, as they are stored and managed by the operating system [17].

The DPAPI offers two types of encryption keys: user specific and system credentials. Both key types are managed securely by the OS, either tied to a user account or to the system itself. In our approach, we employ the user-specific method, which ties encryption to a specific user’s login credentials. This requires the same user to be logged in to successfully decrypt their own data. If the system credentials were used instead, any process or other users running on the system could potentially decrypt any user’s data, which could lead to security issues. Therefore, using the user-specific approach ensures that each user’s data are protected and can only be accessed by the same user.

### 4.3. Data Integrity

The DPAPI handles data integrity through a hashing mechanism, ensuring that the data remain intact and unaltered during storage and retrieval. Throughout the encryption and decryption process, the DPAPI performs integrity checks using cryptographic hashes. If the hash values do not match, the DPAPI can determine that the data have been altered or corrupted and will not return the decrypted data. In our developed program, we utilize the error codes produced by DPAPI functions to identify unauthorized access and potential tampering. In such cases, our application will not return any data.

## 5. Security Analysis

The data protection API (DPAPI) provides robust security against attacks that aim to reverse engineer or intercept encrypted data. Its high level of security is demonstrated by its use in Microsoft Edge, in which it securely encrypts sensitive information such as passwords and credit card numbers when they are saved [18]. Even if an attacker gains local administrator rights and accesses the encrypted data on the local machine while the user is not logged in, the DPAPI is designed to prevent decryption.

One of the key advantages that led us to adopt the DPAPI is its effective management of encryption keys. However, it also has a significant limitation: any application running on the same device and under the same user account as the protected data can potentially access that data if the application is aware of the existence of the protection system. To address this vulnerability, the DPAPI offers an entropy feature that adds an extra layer of complexity by incorporating a randomly generated key phrase into the application’s code. This makes it significantly more difficult for an attacker to access the encrypted data, as they would need to conduct a thorough analysis of the application’s code to extract the key phrase.

## 6. Implementation

As proof of concept that supports the intent of this paper, we created a program named “RunRegProtect.exe”. This application uses a special technique known as hooking in which the registry API calls are intercepted. Then, each read from and write to the registry is monitored, and the data are potentially altered.

This section contains an explanation of the program and its inner workings. It was written in C++ since this language has been integral in Windows development for many years. C++ is also on a lower level than other languages such as C# and a bit easier for programs of this type.

The entire project can be opened, edited, and recompiled using Visual Studio. While Version 2022 was utilized, later versions of Visual Studio should also be compatible. The project is open-source and can be found on GitHub at the link https://github.com/eamoruso/UserPrivacyProtect (accessed on 4 August 2024).

### 6.1. Making Registry Calls

The Windows API functions for registry access are rich in functionality. The functions in Table 2 are a small subset, but they represent the functions that, when hooked, meet our needs. To create a registry entry and write data to the registry in the newly created entry, a sequence of calls to RegCreateKeyExW, RegSetValueExW, and RegCloseKey can be employed as shown in Figure 3.

If a registry key already exists, then a call to the API function RegOpenKeyExW followed by calls to RegSetValueExW and RegCloseKey can be made. This process is shown in Figure 4b.

When examining our code for all calls to RegCreateKeyExW and RegOpenKeyExW, it becomes apparent that the read and write permissions are specified as parameters. These parameters are represented as KEY_READ|KEY_WRITE. Once the registry key is created or opened, it allows both writing data to the key and reading data from it. The RegGetValueW function is used to retrieve the data stored in the key. This process is demonstrated in Figure 4a.

### 6.2. Hooking the Registry

Windows offers many features to application developers. Including the underlying systems that support file input/output, graphics, and networking. These are all provided to application developers through API calls.

The Windows architects realized early on that, in special cases, user applications needed to intercept API calls, perform tasks, or manipulate data and then let the Windows API calls perform their tasks normally. When a user application performs this for an API call, it is called hooking. Most developers never use this technique, and most developers are not aware of its existence.

Windows hooking is an advanced technique. For this reason, the demonstration program uses a library named Detours, which is provided by Microsoft and can be used free of charge for non-commercial use [19]. It allows a programmer to monitor and intercept Windows API calls, providing a safe way to implement hooks. This software package can be found at https://github.com/microsoft/Detours (accessed on 4 August 2024) or NuGet within Visual Studio can be used to import it.

The process of hooking the registry essentially inserts code between the caller (which, in this paper, is Microsoft Word, Excel, and PowerPoint) and the base API code. The steps to complete this for each hooked API function are as follows:Create a function with an identical structure and parameters as the base API function. For example, “my_function(par_1, par_2)” is identical to original function called “real_function(par_1, par_2)” we are going to hook.Get the address of the base API in a variable so that our code can call the base API.In the new function call, use the base API after the additional processing is completed with the saved base API address.Call DetourAttach function to insert the new function in the chain. The operating system will now call our function instead of the base API function. When our function has completed its processing, it will in turn call the base API function.

By using the hooked code, our developed software decides whether to encrypt or decrypt data based on a registry key’s name (e.g., Item 1, Item 2, Item 3, …) that is passed by the application (e.g., Word, Excel, PowerPoint) to the registry. To demonstrate the functionalities in our developed program, a total of seven registry functions are hooked to cover the above three Microsoft Office applications, and these essential functions are listed in Table 2.

### 6.3. Registry Encryption

To effectively secure sensitive information, it is important to have a robust encryption method that can be reliably reversed and decrypted. For demonstrating a protocol, a widely used encryption standard known as the data protection API (DPAPI) was employed in this program. The DPAPI offers convenient services for encrypting and decrypting data without the need to manage cryptographic keys manually. For example, either plaintext data are passed to the DPAPI and an obscure protected data BLOB is received back, or the protected data BLOB is passed to the DPAPI and the plaintext data are received back. The encryption and decryption process can be seen in Figure 5.

In our developed program, we created a pair of functions named “CryptProtectData” and “CryptUnprotectData”. Together, these functions perform the encryption and decryption tasks. A reader can easily modify this section of source code to use other cryptographic algorithms such as Data Encryption Standard (DES). The Windows operating system includes most standard encryption methods.

Our program includes a built-in comparison function named “wcsncpm” that holds the registry key names containing the data we will encrypt and decrypt. In our custom application, we compare the key name with “Item”, which is used by Word to store document names created by the user. Any key names other than these will be ignored, leaving the data unencrypted. Readers can easily add new key names to this function to extend the encryption protection to other targeted Windows applications.

## 7. Evaluation

In our evaluation, we illustrate the success and performance efficiency of our custom-developed application, RunRegProtect.exe, in encrypting sensitive user data stored within the registry. It would be best if we could compare the performance with that of existing systems; however, there are no published works or products on real-time encryption of the registry. Therefore, in this section for the performance evaluation, we only show the performance overhead of our proposed approach compared with the performance of the original OS without installing our real-time registry encryption system.

For the proof of concept, we demonstrate the encryption of data generated by Microsoft Office applications, particularly Word, into the user’s designated registry keys. Our developed program supports Office 2019 through the latest Office 365 version 2401. These keys comprise filenames assigned by the user when saving their documents in Word, spreadsheets in Excel, and presentations in PowerPoint. Using Word as an example, Figure 6 depicts all files created by the current user in the registry (using the automated script shown in Algorithm 2), displayed as plain text.

To produce our results, we utilized a virtual machine (VM) hosted on a Microsoft Hyper-V server running Windows 11 and Microsoft Office 365. Our remote access to the environment was achieved using remote desktop. Additionally, it is important to note that the development and testing of our application were carried out on a Parallels VM operating Windows 11 and Office 2019 Professional Plus. Although we used a later version of Office for development, both versions shared the same registry storage structure, resulting in our custom-developed application performing identically across environments. The specifications of the evaluation system’s hardware and software are outlined in Table 3.

### 7.1. Experiments

To assess the effectiveness and reliability of our application in the context of Word, we conducted a series of experiments utilizing Macro Record [20], a tool capable of recording user mouse and keyboard interactions and providing playback functionality to replicate these actions up to a specified extent. Furthermore, we crafted a Word macro file, illustrated in Algorithm 2, to help automate our simulation. The macro was developed in Visual Basic for Applications (VBA), which facilitates the automation of repetitive tasks and data processing functions. Specifically, we leveraged these capabilities to generate a file stamped with the current date and time, subsequently saving it in the user’s temp directory. This macro file (e.g., template.docm) is executed several times by Macro Record to continuously create new files, testing our application’s performance and accuracy during both encrypted and non-encrypted processes.
**Algorithm 2: MS Word Macro template used to automate experiment****1** **Sub AutoOpen()**2        Dim strFileName As String3        Dim rngHeader As Range4        # Create a header that says “RegRunProtect”5        Set rngHeader = ActiveDocument.Sections(1).6        Headers(wdHeaderFooterPrimary).RangerngHeader.Text = “RegRuProtect”7        # Specify the file name with today’s date and current time8        strFileName = “C:\temp\file_” & Format(Now, “mm-dd-yyyy_HH-MM-SS”) & “.docx”9        # Save and close the document with the specified file name10       ActiveDocument.SaveAs2 FileName:=strFileName11       ActiveDocument.Close**12** **End Sub**

Lastly, we recorded the encryption and decryption processing times for each simulated iteration and saved the results to a comma-separated value (CSV) file. The results of this file were then used to demonstrate the efficiency of our application in the next section.

### 7.2. Results

Our experimental results demonstrate the successful encryption of registry entries containing users’ private information, specifically the location and name of the document. In Figure 6, we present an example of normal registry entries created by Word to store filenames that the user has generated. After running Word with RunRegProtect.exe, our application encrypted those filenames, as depicted in Figure 7.

### 7.3. Performance

During our performance assessment, we found that our application did not significantly degrade or slow down the user experience. To collect our results, we recorded the human operations involved in our simulation by capturing mouse and keyboard events utilizing Macro Record [20]. This operation entailed creating, modifying, saving a Word document with a designated filename and subsequently closing the file. Our experimental script replayed the recorded operation 45 times to obtain the average time required for this operation under both scenarios: word document operations without protection and with protection enabled. This enabled us to compare the application’s timing performance under both conditions.

To facilitate data analysis, we transferred the collected information to a spreadsheet and constructed the graph depicted in Figure 8. Our analysis revealed that the average time required for the process without protection was 15.218 s. Enabling the RunRegProtect.exe feature added a mere 0.051 s to the overall processing time, resulting in an average of 15.269 s. In other words, our proposed security mechanism only adds about 0.34% overhead in terms of runtime. The standard deviation for the process without protection was 0.2699 s compared to 0.3405 s with protection enabled. Overall, our results suggest that the application’s performance is not significantly impacted by enabling the RunRegProtect.exe feature.

## 8. Conclusions

This research presents a new paradigm of security approach to safeguarding sensitive information within the registry. By combining software-based encryption with advanced hooking techniques, our proof-of-concept application successfully protects user privacy and security without requiring modifications to the operating system or installed software. The integration of the Microsoft data protection API (DPAPI) further enhances the security of our solution, making it exceedingly difficult for malicious actors or applications to obtain a user’s encryption key and then access the protected registry information. Through several simulations, we demonstrated the flawless and accurate encryption and decryption capabilities for a specific user’s private registry information. Notably, our application’s performance overhead was minimal, with a mere 0.34% impact on system performance when hooking and encrypting registry data. This research contributes significantly to the development of more secure and private computing environments, ultimately empowering individuals and organizations to protect their sensitive data with confidence.

## Figures and Tables

**Figure 1 sensors-24-05106-f001:**
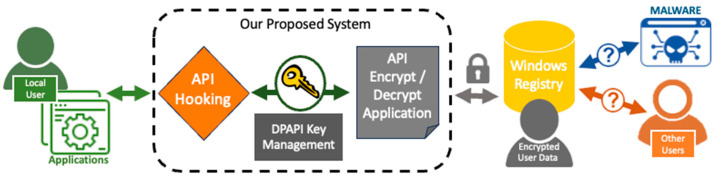
Overall diagram highlighting our approach. The dotted box denotes our proposed system responsible for intercepting and encrypting/decrypting communication between a user’s applications and registry. Malware and other users cannot obtain the user’s registry data since they are encrypted.

**Figure 2 sensors-24-05106-f002:**
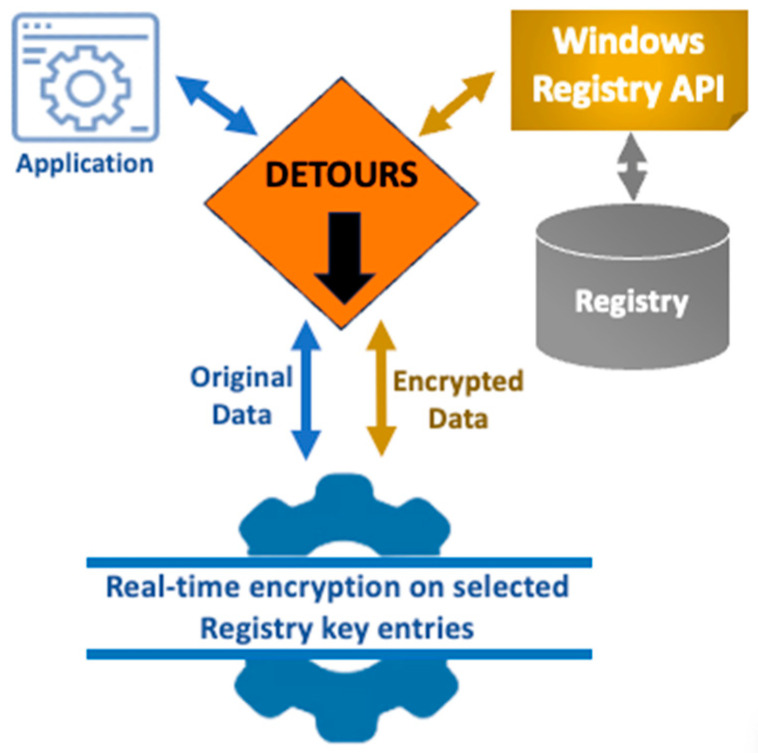
Detours function to hook and capture an application’s registry API calls and modify selected registry key data.

**Figure 3 sensors-24-05106-f003:**
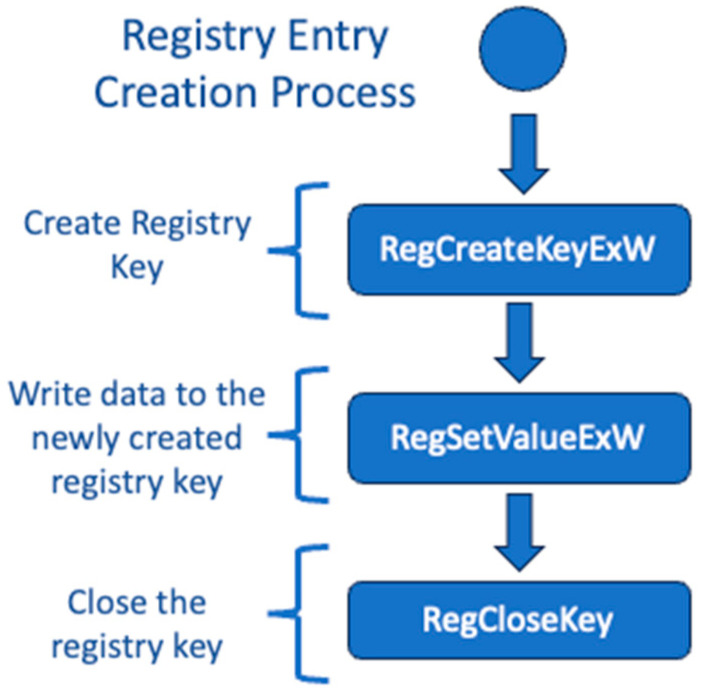
Creating and saving data to a new registry key.

**Figure 4 sensors-24-05106-f004:**
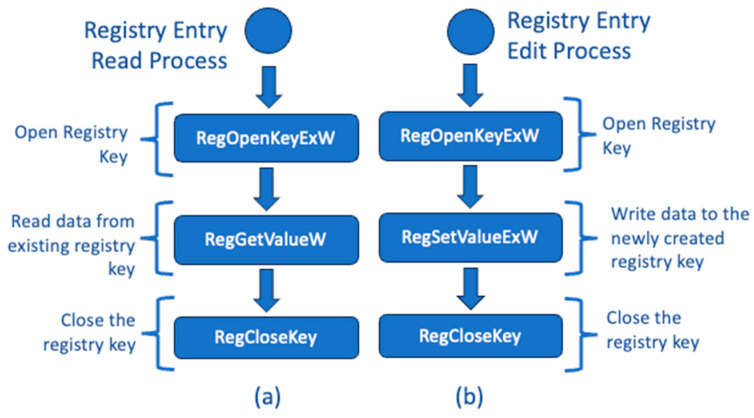
(**a**) Reading data from a registry key entry. (**b**) Creating and saving data to an existing registry key.

**Figure 5 sensors-24-05106-f005:**
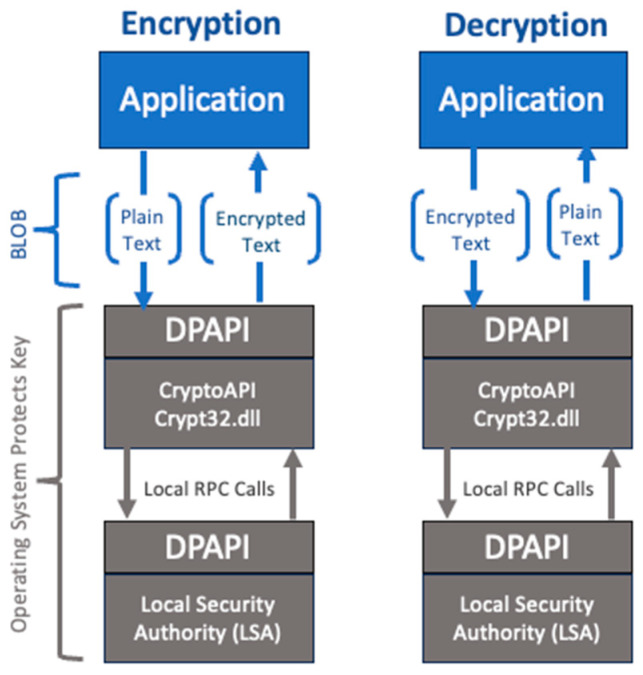
DPAPI encryption and decryption process where user’s encryption key is managed by the Operating System’s Local Security Authority (LSA) through local remote procedure calls (RPCs).

**Figure 6 sensors-24-05106-f006:**
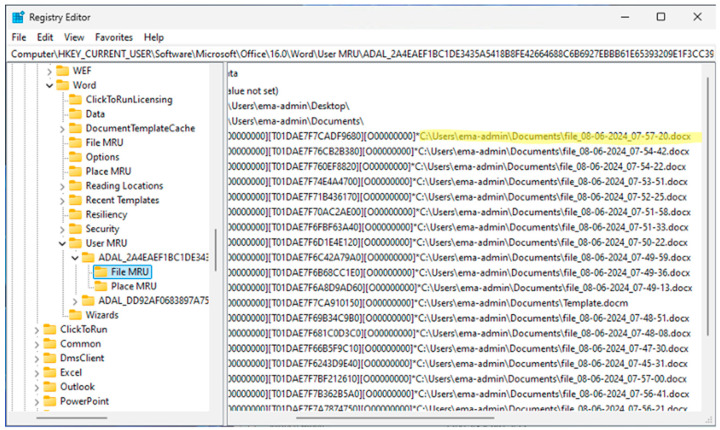
Registry entry listing all Word files the user created on the system. The highlighted entry specifies the location and name of the user’s file.

**Figure 7 sensors-24-05106-f007:**
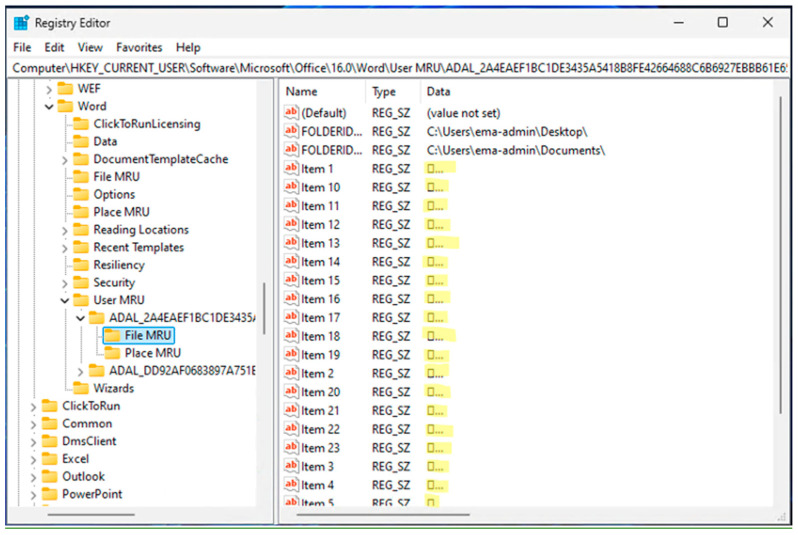
Registry entries of Word documents successfully encrypted by RunRegProtect.exe. The highlighted items are encrypted. In this example, User-A is trying to access User-B’s (encrypted) registry information using regedit.exe. By design, regedit will display the encrypted data as a box with trailing periods.

**Figure 8 sensors-24-05106-f008:**
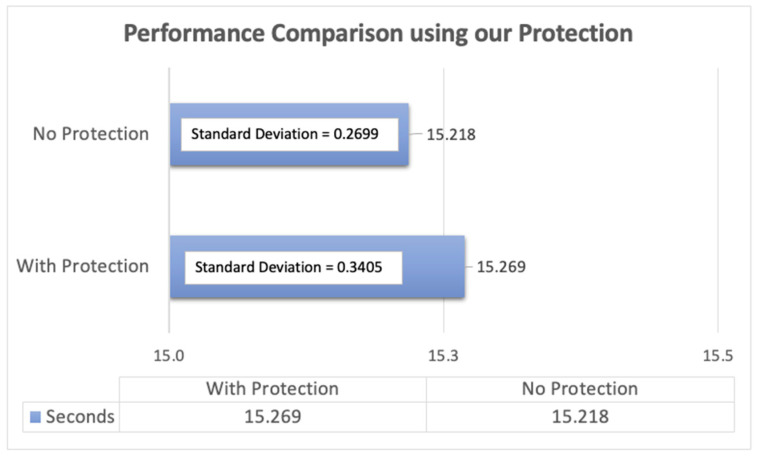
Performance comparison between using our protection and no protection.

**Table 1 sensors-24-05106-t001:** List of abbreviations.

Abbreviation	Full Form
API	Application programming interface
DPAPI	Data protection application programming interface
OS	Operating system
ACL	Access-control list
FDE	Full-disk encryption
VM	Virtual machine
AES	Advanced encryption standard

**Table 2 sensors-24-05106-t002:** Registry API functions hooked by our developed program RunRegProtect.exe.

Registry Function	Description
RegOpenKeyExW	Opens the specified registry key.
RegCreateKeyExW	Creates the specified registry key.
RegSetValueExW	Sets the data and type of a specified value under a registry key.
RegQueryValueExW	Retrieves the type and data for the specified value name associated with an open registry key.
RegEnumValueExW	Enumerates the values for the specified open registry key. The function copies one indexed value name and data block for the key each time it is called.
RegEnumKeyExW	Enumerates the subkeys of the specified open registry key. The function retrieves information about one subkey each time it is called.
RegCloseKey	Closes a handle to the specified registry key.

**Table 3 sensors-24-05106-t003:** Hardware specifications for evaluation system.

Type	Description
System Hardware	Dell PowerEdge R440 Intel Xeon Silver 2.1 GHz Processor
Operating System	Windows 11 Enterprise Version 23H2 (OS build 22631.3296)
Microsoft Office Suite	Office 365 MSO 64-bit for Enterprise Version: 2401 Build 16.0.1731.20290

## Data Availability

Data are contained within the article.

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
