# Peer review of "User Privacy Protection via Windows Registry Hooking and Runtime Encryption"

_sensors, 2024, doi:10.3390/s24165106_

Round 1

Reviewer 1 Report

Comments and Suggestions for Authors

Some issues in this manuscript need to be addressed:

1.       The abstract is poorly written as it does not summarize the key points of the article effectively. It is recommended that it be rewritten to include a summary of the results as well. Please refer to point no. 7 given below.

2.       In the Literature Review section, include a comparison table discussing cybersecurity research. Highlight the strengths, weaknesses, and applications of existing published studies.

3.       In Section 3, the authors need to discuss recently introduced threat models, such as https://doi.org/10.1049/gtd2.13103 and https://doi.org/10.1049/blc2.12081. The authors can also incorporate the mathematical modeling presented in these works, just by adding references.

4.       It is recommended that the pseudo code for the proposed scheme be included in the paper, along with a detailed discussion, in Section 4.

5.       Sections 4.1 and 4.2 require more explanation and should include additional mathematical modeling refer to point 3.

6.       The authors need to add the simulation parameters and their values in the simulation section (Section 5).

7.       The authors must include simulation graphs comparing the proposed scheme to existing studies in Section 5.

8.       It is recommended that a motivation paragraph be added to explain how and why the proposed methods are adequate.

9.       Summary of the paper need to be revised based on the comparison analysis. 

Author Response

Comment 1: The abstract is poorly written as it does not summarize the key points of the article effectively. It is recommended that it be rewritten to include a summary of the results as well. Please refer to point no. 7 given below.

Response 1: We rewrote the abstract to make the motivation concise and clear. In addition, we summarized the key contribution points and ideas in the abstract to make it self-contained and easier to understand.

Comment 2: In the Literature Review section, include a comparison table discussing cybersecurity research. Highlight the strengths, weaknesses, and applications of existing published studies.

Response 2: Through our research, we cannot find any peer-reviewed research that conduct real-time encryption/decryption on Windows registry like what we proposed in this paper. In our revision, we revised related work section and added multiple new reference (lines 81-99). For related work on protecting Windows registry, we categorized them into three classes of methods (lines 84-90) and summarized related work in the most related two areas: “disk encryption” and “access control list”. 

Comment 3: In Section 3, the authors need to discuss recently introduced threat models, such as https://doi.org/10.1049/gtd2.13103 and https://doi.org/10.1049/blc2.12081. The authors can also incorporate the mathematical modeling presented in these works, just by adding references.

Response 3: :  Thanks for pointing out these two important research papers. In our revision, we have added these two references to the threat model (Section 3) as [12][13] (line 150). 

Comment 4: It is recommended that the pseudo code for the proposed scheme be included in the paper, along with a detailed discussion, in Section 4.

Response 4: Thanks for this great suggestion. In our revision, in the ‘Proposed Approach’ section (Section 4), we have added the overall system diagram (Figure 1) and its explanation. In addition, we have added the pseudo code of our proposed approach (Algorithm 1). 

Comment 5: Sections 4.1 and 4.2 require more explanation and should include additional mathematical modeling refer to point 3.

Response 5: We have rewritten the entire Section 4 and added a lot of new content and explanations. We added the Figure 1 to show the overall system architecture, and Algorithm 1 of the high-level pseudo code to illustrate the proposed system’s operation. 

Comment 6: The authors need to add the simulation parameters and their values in the simulation section (Section 5).

Response 6: Thanks for this great suggestion. In our revision of the evaluation section (new Section 7), we added detailed explanation of the experiment setup at Section 7.1 and added Algorithm 2 to show the script we programmed to automate the Word file generation and registry key adding operation. The simulation parameter (script replayed the recorded operation 45 times) used in obtaining our evaluation results shown in Figure 8 is explained in the first paragraph of Section 7.3 (line 424-432).   

Comment 7: The authors must include simulation graphs comparing the proposed scheme to existing studies in Section 5.

Response 7: So far, we haven’t been able to find any existing studies conducting real-time encryption/decryption on Windows registry information. Thus, we can only compare the performance of our approach with the original OS system that does not have our system installed. In our revision, we have added the first paragraph (line 365-371) in the evaluation section (Section 7) to explain this point. 

Comment 8: It is recommended that a motivation paragraph be added to explain how and why the proposed methods are adequate.

Response 8: We have revised the introduction section (Section 1) and added more explanation of the motivation to make it clearer (line 44-67).

Comment 9: Summary of the paper need to be revised based on the comparison analysis.

Response 9: We rewrote the summary of paper (Section 8) for better flow and to reflect our analysis. 

Reviewer 2 Report

Comments and Suggestions for Authors

The manuscript describes an approach for preserving user privacy and data confidentiality in the Windows registry. While the proposed approach is interesting, there are several areas where the manuscript could be improved to increase clarity, completeness, and scientific rigor:

1.There are spelling and typographical errors that must be corrected. For example, the word "discreet" in the abstract should be changed. Section 4.2 appears to repeat itself, which should be addressed to improve readability and coherence.

2.The discussion of "Using Encryption" and "Using Access Control" methods in the related work section does not include adequate citations to scientific works that have used these methods. The authors must provide references to previous studies that used these methods and compare them to the approach presented in this paper.

3.A broad diagram of the proposed approach should be included in the paper. This diagram should describe the adopted methodology and its various components in detail. A visual representation can help readers understand the workflow and structure of the proposed approach.

The characteristics mentioned on page 170 for the prototype design features must be clearly explained.

4.While the paper emphasizes data confidentiality, it does not address data integrity, which is essential in the context of this research. The proposed technique, which allows for selective modification of registry values, could be used by threat actors to compromise data integrity. The authors should explain how their approach addresses data integrity and suggest additional safeguards against attacks on integrity. 

5.The decision to use Microsoft's Data Protect API (DPAPI) is not justified. The authors should provide a schematic explanation of the sections "Securing with Encryption" and "Security Analysis" as well as a discussion of other competing solutions. Justifying DPAPI's advantages over other alternatives will strengthen the case for its selection.

6. The experimental evaluation in this paper is limited. The authors should simulate attacks on the proposed approach to demonstrate its robustness and effectiveness.

7. Comparisons with other existing solutions or published works are required to fully evaluate the proposed solution's effectiveness.

8. Perspectives are missing. 

Author Response

Comment 1: There are spelling and typographical errors that must be corrected. For example, the word "discreet" in the abstract should be changed. Section 4.2 appears to repeat itself, which should be addressed to improve readability and coherence.

Response 1: We revised the abstract completely and made sure that the abstract is clear and well-structured showing our research goals and contributions. For the original Section 4.2, we revised and expanded it as the new Section 5 (Security Analysis) to explain why the Microsoft DPAPI is secure for achieving the user key management task in our proposed system. 

Comment 2: The discussion of "Using Encryption" and "Using Access Control" methods in the related work section does not include adequate citations to scientific works that have used these methods. The authors must provide references to previous studies that used these methods and compare them to the approach presented in this paper.

Response 2: We revised both sections and added more citations related to scientific works used in these two methods. 

Comment 3: A broad diagram of the proposed approach should be included in the paper. This diagram should describe the adopted methodology and its various components in detail. A visual representation can help readers understand the workflow and structure of the proposed approach. The characteristics mentioned on page 170 for the prototype design features must be clearly explained.

Response 3: Thanks for this great suggestion. In our revision, we have created an overall system diagram as Figure 1 and expanded on the explanation of the system design features to make them clearer and easier to understand (line 164-210). 

Comment 4: While the paper emphasizes data confidentiality, it does not address data integrity, which is essential in the context of this research. The proposed technique, which allows for selective modification of registry values, could be used by threat actors to compromise data integrity. The authors should explain how their approach addresses data integrity and suggest additional safeguards against attacks on integrity. 

Response 4: Thanks for pointing out this overlooked point by us. In our revision, we have added a new subsection (Section 4.3) to explain how we handle data integrity. 

Comment 5: The decision to use Microsoft's Data Protect API (DPAPI) is not justified. The authors should provide a schematic explanation of the sections "Securing with Encryption" and "Security Analysis" as well as a discussion of other competing solutions. Justifying DPAPI's advantages over other alternatives will strengthen the case for its selection.

Response 5: In our revision, we explained more details and security features of DPAPI in Section 4.2, 4.3 and the new Section 5.  These revisions should provide better justification why we choose DPAPI for the user key management in our proposed system. 

Comment 6: The experimental evaluation in this paper is limited. The authors should simulate attacks on the proposed approach to demonstrate its robustness and effectiveness.

Response 6: We rewrote most content in this evaluation section (Section 7) to have a better and clearer explanation. Figure 7 shows what an attacker could obtain when he or she has the access to the registry data. It demonstrates that because of the deployed real-time encryption by our program, an attacker can only see the key entries with encrypted key values. This illustrates the effectiveness of our approach when another user tries to read encrypted registry data.

Comment 7: Comparisons with other existing solutions or published works are required to fully evaluate the proposed solution's effectiveness.

Response 7: So far, we haven’t been able to find any existing studies conducting real-time encryption/decryption on Windows registry. Thus, we can only compare the performance of our approach with the original OS system that does not have our system installed. In our revision, we have added the first paragraph (line 365-371) in the evaluation section (Section 7) to explain this point. 

Comment 8: Perspectives are missing. 

Response 8: In the revision, we try our best to explain our research from different perspectives in multiple places. In the Section 3, we expanded explanation of threat model to better explain what attacks our proposed approach can deal with. In the evaluation section 7, we showed the output of attack against our system in Figure 7, demonstrating what information (key values are encrypted and unreadable) could be obtained when an attacker gets access to the registry data. Section 7.3 shows the performance overhead when implementing our proof-of-concept code compared with the original unprotected system. 

Round 2

Reviewer 1 Report

Comments and Suggestions for Authors

The authors have addressed our comments in the revised version. Therefore, we accept this paper in its current form.

Author Response

Thanks so much for this reviewer's review effort and approval of our first-round revision. We made some more editings in this second-round revision (yellow-highlighted) and also added a new table (Table 1, line 81) showing the list of abbreviations used in this paper. 

Reviewer 2 Report

Comments and Suggestions for Authors

Most of the comments have been considered

Author Response

Thanks so much for this reviewer's review effort and approval of our first-round revision. We made more editings in this second-round revision (yellow-highlighted) and also added a new table (Table 1, line 81) showing the list of abbreviations used in this paper.